# Ground-Penetrating Radar and Photogrammetric Investigation on Prehistoric Tumuli at Parabita (Lecce, Italy) Performed with an Unconventional Use of the Position Markers

**Raffaele Persico** [1,*], **Emanuele Colica** [2], **Tiziana Zappatore** [3], **Claudio Giardino** [3] and **Sebastiano D'Amico** [2]

1 Department of Engineering of the Environment, University Campus, University of Calabria, 87036 Rende, Italy

2 Department of Geoscience, University Campus, University of Malta, MSD 2080 Msida, Malta; emanuele.colica@um.edu.mt (E.C.); sebastiano.damico@um.edu.mt (S.D.)

3 Department of Cultural Heritage, University of Salento, Via Dalmazio Birago 64, 73100 Lecce, Italy; tiziana.zappatore@studenti.unisalento.it (T.Z.); claudio.giardino@unisalento.it (C.G.)

* Correspondence: raffaele.persico@unical.it

**Abstract:** In this contribution, we propose ground-penetrating radar (GPR) investigation performed close and on some prehistoric tumuli, locally called "*piccole specchie*", in the countryside around the town of Parabita (Lecce), within the Salento peninsula (southern Italy). In order to perform the GPR investigation on the tumuli, an unconventional method of data acquisition was exploited, involving, consequently, some non-conventional data processing steps. Photogrammetric survey was also performed, and 3D digital models of the prehistoric tumuli were created. The investigations have revealed some anomalies under two out of three investigated tumuli, which were interpreted as prehistoric tombs.

**Keywords:** ground-penetrating radar; position markers; prehistory; photogrammetry

## 1. Introduction

Although investigation on cultural heritage is one of most classical applicative fields of ground-penetrating radar (GPR) prospecting [1–7], to our knowledge, in southern Italy, there had been no previous GPR investigation on structures similar to the tumuli at hand, also called *piccole specchie*.

Tumuli are stone heaps made of soil and stones, raised over a single or multiple grave. The word tumulus in Latin means "mound" or "small hill". Tumuli are also known as barrows, cairns, burial mounds or kurgans. Tumuli can be circular or elongated in shape, and they are spread all over prehistoric Europe and Asia, starting from the Neolithic period. Many of them can be ascribed to the Bronze Age, but there are tumuli dating back to the Iron Age and even to the classical period and to the medieval time.

Tumuli are inhomogeneous structures, and they do not have a particular design because they are composed of stones, which are simply accommodated one on top of the other to create a mound. At Parabita, the height of the tumuli ranges between one and two meters. In this paper, we have prospected three tumuli, labeled $T_1$, $T_2$ and $T_3$ (Figure 1). Their location is given in Figure 1, with a Google Earth satellite view of the investigated area and its geographic location near a cave known as "Grotta delle Veneri", frequented since the Paleolithic Age.

As can be understood from Figure 1, the structure of the tumulus hinders the use of the metric wheel of a GPR system. Therefore, we have made use of the user-stacking acquisition mode, where the antennas are pulled and/or pushed along the observation line (i.e., the usual trolley is dismounted). As is well known, in this acquisition mode, position markers are needed. The classical use of the position markers is performed by recording the marker points while the GPR passes by them. In particular, when the antenna moves across

the marker position, the human operator "picks up" the marker on the screen (usually a touch screen) of the personal computer (PC), driving the data acquisition. In particular, we have made use of the K2 FASTWAVE acquisition software, which works in this way. The classical user-stacking modality, however, implies a physical delay between the moment when the marker is identified and the moment when it is gathered on the screen. Moreover, on a sunny day, some marker positions can be missed simply because the screen of the computer becomes hardly visible. However, even if a few marker points are missed, on condition that the human operator keeps his/her walking velocity sufficiently constant, the missed marker points can be estimated and added during the post-processing phase. This can be performed on the basis of the number of nominal marker positions that should have been gathered and on the basis of the wideness of the apparent intervals between any two consecutive, actually gathered marker points. Of course, the lacking marker points will be added in the middle of those intervals, which appear anomalously larger with respect to the other ones.

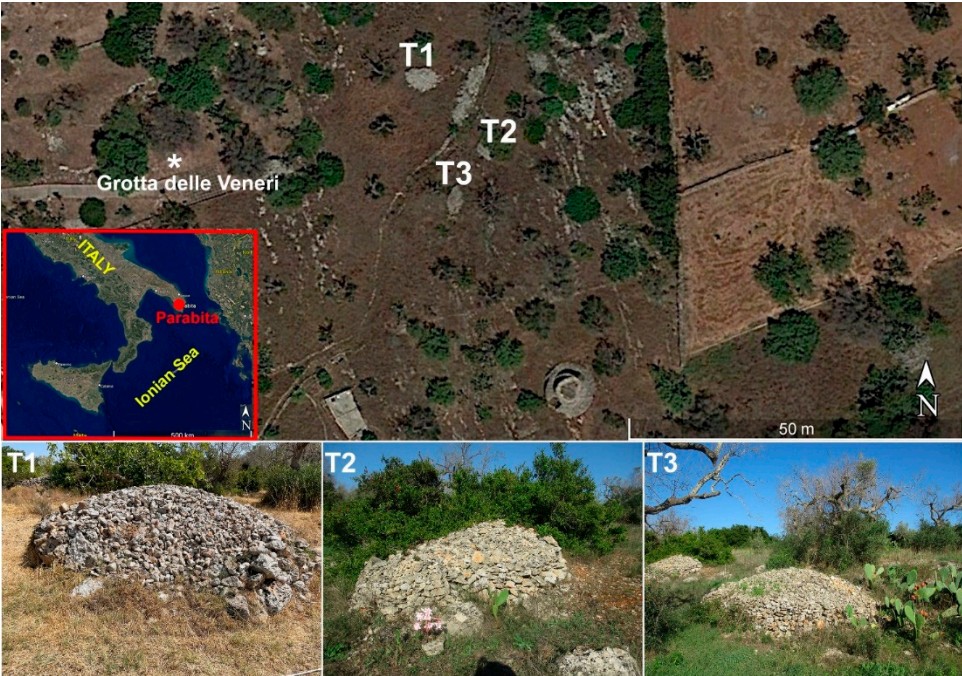

**Figure 1.** Top panel: The geographical position of the site with the area of the burial mounds (tumuli) and of the "Grotta delle Veneri" (Latitude 40.053° N; Longitude 18.198° E). Bottom panels: Photographs of the three investigated tumuli. From the left: $T_1$, $T_2$ and $T_3$, respectively.

In spite of these drawbacks, the classical user-stacking acquisition mode is user friendly and provides satisfying results for many applications [8]. The precision available with regard to the pick-up of the marker positioning is difficult to predict, but in our experience, it can be estimated in the order of 20–30 cm. In case of a missed marker, however, the available precision depends on the distance between any two consecutive marker positions established in the field, and on average, it is expected to be worse than 20–30 cm.

In the case scenario proposed here, the observation lines are only a few meters long, and we had the practical exigency to place marker points at any 50 cm distance from each other. Moreover, the topography of the tumulus added some supplementary difficulty in the correct gathering of the marker positions, due to the precarious equilibrium of the human operator on the tumulus. Consequently, we could not accept a precision of 20–30 cm with regard to the positioning of the makers, and neither could we accept the possibility of missing any marker point. Therefore, we decided to adopt the strategy of stopping the instrument for some seconds at each marker point. In this way, the marker positions can be recognized in the post-processing phase because they appear as horizontally constant

pieces of data. In fact, since the antenna is kept still at the marker point, the instrument launches and gathers several times the same signal vs. time (also called Ascan [9]). As will be shown, this results in a sequence of horizontal belts piled on each other within the comprehensive matrix of data (i.e., the data vs. abscissa and the return time, also called Bscan [9]). Moreover, by adopting this data acquisition mode, it is virtually impossible to miss a marker point, and the human operators can visually check at each stopover the precision of the positioning of the instrument at the current marker point. In the case at hand, we observed a precision of the order of 2–3 cm with regard to the gathered marker positions.

Data acquisition in user-stacking mode with stopovers of the antenna had been experimented with in [10]. In that case, it was applied as an exercise on the floor of a corridor of no applicative interest. However, that situation allowed a comparison with data gathered with the metric wheel. In the present paper, the stopover acquisition mode is, for the first time (to our knowledge), applied to a real-case scenario.

In the next section, we will provide an insight into the archaeological context of the tumuli in the geographic area at hand (which is called "Salento"). In Section 3, the processing of GPR data, as well as the photogrammetric survey, will be described. In Section 4, the results achieved from GPR data taken on the tumuli will be shown, whereas in Section 5 the results of GPR measurements performed within an area close to tumulus $T_1$ will be shown. Conclusions will follow.

## 2. The Archaeological Context of the Tumuli

The proto-historic tumuli, belonging to the burial ground of Parabita–Madonna della Grottella, are completely new to the archaeological studies. Only 43 m from the graves, there is the relevant prehistoric site of the Grotta delle Veneri ("Venuses Cave"); Grotta delle Veneri is a cave mostly frequented during the Paleolithic and the Neolithic, but there is also some evidence of it belonging to the Copper and the Bronze Age [11,12].

The Bronze and Iron Age settlement of Cava Stefanelli is close to the graves, at 376 m from the tumuli. Mycenaean sherds were found at Cava Stefanelli; they are evidence of the connections of the Parabita area with the Aegeans in the Middle Bronze Age [13]. The main objective of studying the Parabita graves is to expand the scarce knowledge currently available on protohistoric tumulus-like burials in the Apulian area.

The Salento peninsula is the southern part of the Apulia region, between the Ionian Sea to the west and the Adriatic Sea to the east. Due to its geographical position, relatively isolated from the rest of the Italian peninsula but very close to the Albanian and Greek coasts, Salento often had its own peculiarities, especially in the pre-protohistoric periods.

In Salento, the most frequent prehistoric funerary structures are dolmenic cist tombs built with slabs of local limestone, covered by heaps of irregular limestone stones with or without the addition of terrain. These tumuli are known, in archaeological literature, as *specchie*, or, more precisely, *piccole* (small) *specchie*. Unfortunately, modern archaeological investigations on these peculiar structures are very limited. In addition, due to their visibility, the tombs frequently suffered violations in the past. The tumuli of Parabita belong to the so-called small *specchie*. These megalithic structures are documented in the area between Vanze and Acquarica (Lecce), where some of them were excavated in the first half of the last century [14,15]. Generally, the stone heaps have a diameter between 15 and 30 m.

According to [14], the small *specchie* were used for individual burials. The grave furniture consisted mostly of impasto vessels. The study of the recovered pottery allowed the dating of these monuments from Vanze-Acquarica to the beginning of the Middle Bronze Age, to the so-called proto-Apennine *facies,* and more precisely, to an early moment of this *facies* (c.a. 1700–1500 B.C.).

Other, more recent data come from four dolmenic tombs inside the *Specchia* Artanisi [16] in Ugento (Lecce), excavated in early 21st century. This *Specchia* is formed by at least two adjoining tumuli of 30 and 45 m in diameter. The megalithic structures are

very similar to those of Vanze; the *Specchia* Artanisi has single and multiple depositions. Proto-Apennine-type pottery with archaic typological features was recovered in the grave furniture of this *Specchia* in Ugento. However, in addition, a small bronze dagger was also found, apparently dating back to the Early Bronze Age [17].

The tumuli (or small *specchie*) of Salento studied thus far seem to have been built and used in the first half of the second millennium B.C. (about 2000–1500 B.C.). The Parabita tumuli were probably the burial place of the Bronze Age communities that lived in adjacent areas, such as those living at the Cava Stefanelli settlement.

## 3. Photogrammetric Survey and GPR Investigation

The three tumuli studied in this paper were surveyed from the ground using a Canon EOS 1300D camera equipped with an 18 mm focal length lens. About 90 photographs for each tumulus were acquired in manual mode, trying to obtain the recommended overlap values of at least 80% forward overlap and at least 70% for side overlap between two or more consecutive shots [18]. In Table 1, the main characteristics of the exploited camera are exposed.

**Table 1.** Camera technical specs.

|  | Canon 1300D |
| --- | --- |
| Sensor | APS-C 18 megapixel CMOS (22.3 × 14.9 mm) |
| Image format | 5184 × 3456 pixels |
| Lens | EF-S 18–55 mm f/3.5–5.6 IS II |
| Focal length | 18 mm |
| Auto focus | Yes |
| Aperture | F10 (fixed) |
| Output format | RAW + JPEG |

The acquired images were then processed using Agisoft Metashape [19] digital photogrammetry software, which integrates algorithms deriving from computer vision, such as structure-from-motion (SfM). This system allows the estimation of the 3D position of points represented in multiple images, reconstructing the geometry of the object represented and the position of the camera, even if internal orientation parameters were not defined. The photogrammetric processing followed is the typical pipeline, which involves a series of consecutive steps, as indicated in Figure 2 [20–22]. The first step of the photogrammetric workflow is the import of images into the photogrammetric software, where they will be processed. The successive step, called camera alignment, allows for the automatic orientation of cameras and images in space. A sparse point cloud is then created, which forms a 3D point cloud with scattered points. In the next phase, the low-density point cloud is thickened by increasing the number of points, and a dense point cloud is generated. Starting from this one, a continuous surface is reconstructed made up of polygons whose vertices are the points of the dense cloud. The previous step is called mesh reconstruction and, once the texture is applied to this model, the result is a textured 3D model.

The 3D models were then scaled using ground control points (GCPs) with measurement scales, positioned near the tumuli, at the same time as the images were acquired. Finally, the results obtained from the photogrammetric processing allowed us to extract a profile from the digital elevation models (DEMs) of the three tumuli, coinciding with the position of the measuring tapes positioned on the tumuli for the GPR prospecting.

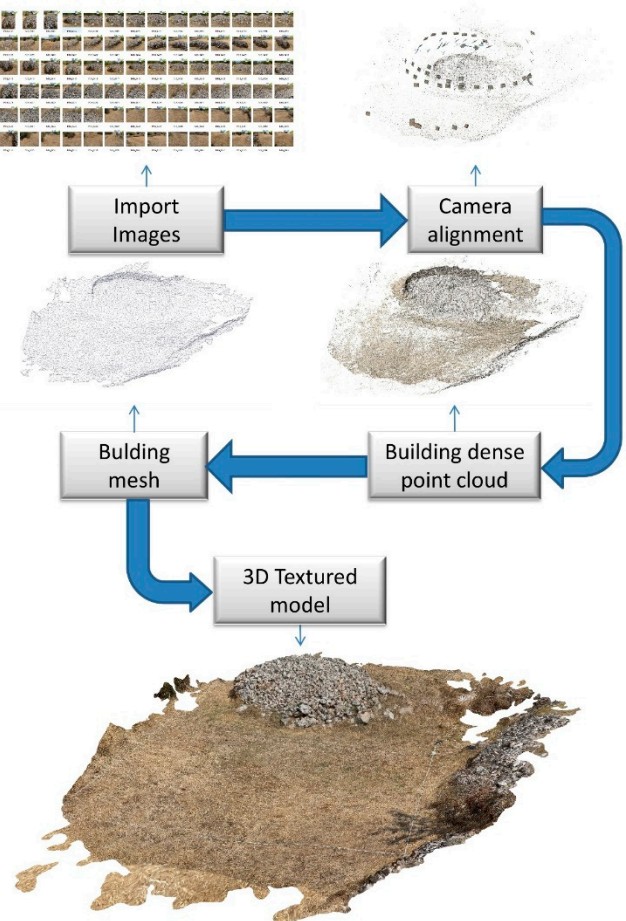

**Figure 2.** Photogrammetric workflow using Agisoft Metashape software [17,18].

Afterward, the three tumuli were crossed with a GPR Hi-mode system manufactured by IDSGeoradar s.r.l. (https://idsgeoradar.com/, last access on 21 January 2022), equipped with a dual antenna at nominal central frequency 200 and 600 MHz, respectively. The measurement line was 4.5, 2.5 and 3.5 m long with regard to tumuli $T_1$, $T_2$ and $T_3$, respectively, and, as said, the data were gathered in user-stacking mode and by stopping the antenna at all the marker points, also including the starting point and the final point. The GPR acquisition parameters were set with a time step of 0.125 ns for the antennas at 600 MHz and 0.25 ns for the antennas at 200 MHz, and with 1024 time samples in both cases. A slowing factor 10 was assigned to the launching of the time pulses (this corresponds to fixing the time interval between the transmission of two consecutive electromagnetic pulses). We chose this setting on a heuristic basis, looking at the velocity with which the data progressively filled up the screen.

We will now show the main operational steps leading to the processed Bscan gathered on tumulus $T_1$ at 600 MHz. A fully analogous procedure was followed with regard to tumuli $T_2$ and $T_3$, for which only the final result will be shown.

In the first step of Figure 3, the raw data at 600 MHz gathered on $T_1$ are shown. The software exploited for the post-processing is the Reflexw (https://www.sandmeier-geo.de/reflexw.html, access on 21 January 2022). As can be seen, the raw data are not easily readable at all, and in the case at hand, we see an initial apparent length of the Bscan being much longer than its real value, due to the deactivation of the metric wheel. In Figure 3 (second step), the result achieved after zero timing (at 9 ns) and after background removal on all traces [23], is shown. From the zoom proposed in the third step of Figure 3, we can appreciate several horizontally constant pieces of Bscan, which are relative to the stopovers on the marker points. The parts where the instrument was moving instead correspond to the narrower zones between the dashed-dotted lines. We then applied a gain variable vs.

time-depth (a linear and exponential gain with parameters 0.5 and 2, heuristically chosen after some trials) and a Butterworth filter with lower and upper cut-off frequencies at 70 and 1500 MHz, respectively [24,25]. The Butterworth filtering mitigated the effects of the spurious enlargement of the band of the signal caused by the gain vs. depth, which is a linear but not time-invariant operation, often causing a peak of the spectrum on its low-frequency side in particular. We have also cut the image at the maximum time depth of 70 ns because only noise was visible beyond this time-depth level. The result achieved after these steps is shown in the fourth step of Figure 3.

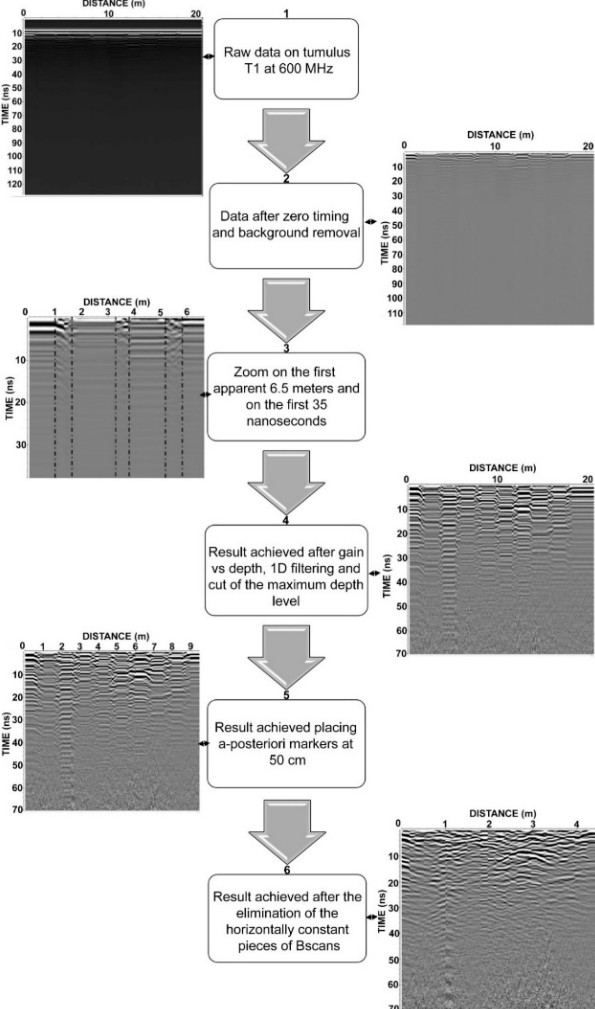

**Figure 3.** Flowchart of GPR processing in the software ReflexW (Reference).

We then placed markers at all the boundary points between flat (horizontally constant) and non-flat zones, plus an initial marker at the beginning of the Bscan and a final marker at the end. We have set a distance between all the imposed markers at 50 cm because this was the real distance between any two consecutive stopovers. The spatial step of the data was interpolated to 1 cm. The result is exposed in the fifth step of Figure 3. The apparent length of the Bscan is now 9.5 m instead of its actual value of 4.5 m.

This occurs because, by construction, we have added to the $n$ (9, in the case at hand) pieces of the "authentic" Bscan, further $n + 1$ (10, in the case at hand) false pieces of Bscan provided by the marker points where the GPR was not moving but notwithstanding was gathering data. In fact, by construction, any stopover point now results in an apparent piece of Bscan, horizontally constant and 0.5 m long. We now only have to remove the horizontally constant parts of the Bscan. Due to the interpolation measurement step set

at 1 cm, this amounts to removing the first 50 traces, retaining the traces from the 51st to the 100th, discharging the traces from the 101st to the 150th, and so on. After this alternate retain–erase procedure, the resulting Bscan is that, which is represented in the sixth step of Figure 3. The result shows some vertical seam effect due to the non-perfect adjacence of the pieces. It is an imprecision that we have mitigated heuristically with trials about the exact boundary points between the true pieces of Bscan and the false ones related to the stopovers. However, its complete elimination was not possible.

## 4. Results Achieved on the Tumuli

The results obtained from the photogrammetric survey are shown in Figure 4. They allowed us to extract a high-precision topographic profile of the tumuli section scanned with the GPR (incidentally, we did not have a differential GPS at disposal). From these profiles, the information of length and relative height were extracted and subsequently inserted in the GPR processing software to correct the Bscans, as shown in Figure 5.

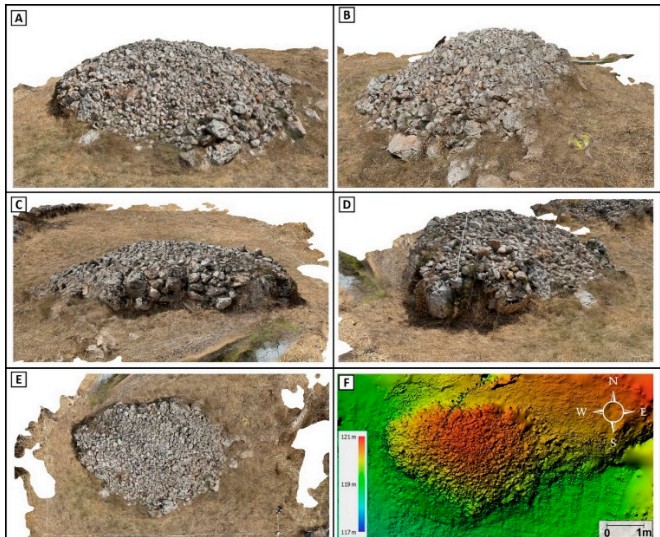

**Figure 4.** Screenshots of the 3D model of tumulus $T_1$ obtained from photogrammetric process with exposure to south (**A**), east (**B**), north (**C**), and west (**D**). Orthogonal view of the tumulus $T_1$ area reconstructed by orthorectification of the images acquired during the photogrammetric survey (**E**). In window (**F**), there is a screenshot of the digital elevation model (DEM), built through photogrammetric processing, in which it is possible to read the absolute elevation values of tumulus $T_1$ above sea level associated with a color scale (see legend).

After applying the topographic correction (we did not have the possibility to account for the tilting of the antennas while moving along the tumulus), we obtained the images displayed in Figure 5, where two clear anomalies are put into evidence by means of a rectangular box. These anomalies are probably the testimony of the ancient funerary use of these structures.

It can be observed, in particular, that the comprehensive length observed in abscissa in Figure 5A is slightly smaller than 4.5 m. In fact, the tape was indeed displaced on the tumulus, and so 4.5 m is the length of the curved path run through. Coherently, its horizontal projection is shorter. Indeed, the functionality of the topographic correction of the Reflexw does not implement this automatically, but it simply pulls up or pushes down the heights of the points that we chose for this purpose, at a parity or resulting abscissas. So, we have worked out the actual length of the horizontal projection of the Bscan through easy trigonometric calculations and have coherently resampled the data a second time. The same procedure was applied to the other two tumuli, and the relative results are shown in Figure 5B (tumulus $T_2$) and 5C (tumulus $T_3$), respectively. Analogous results were achieved from the data at 200 MHz, not shown here for the sake of brevity.

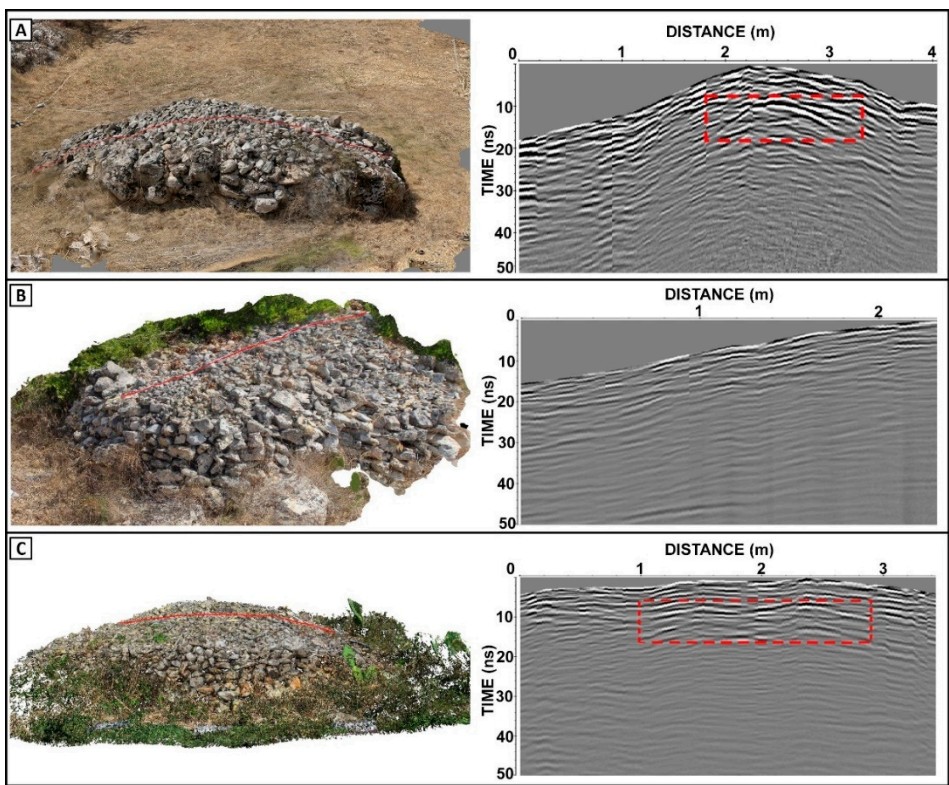

**Figure 5.** (**A**) Elaborated Bscan with topographic corrections derived from photogrammetric models for tumuli $T_1$ (**A**), $T_2$(**B**) and $T_3$ (**C**). On the left-hand side: Virtual reconstructions of the tumuli with the path of the tape followed by the GPR in red. On the right-hand side: The archived reconstruction with tomographic correction. The dashed red lines indicate the main anomalies, interpreted as the top of tombs.

The three presented Bscans were not migrated, essentially because the curvature ray along the measurement line (which is of the same order of the involved wavelengths) rendered the migration result unreliable. For the same reason, indeed, we did not have a precise evaluation of the average propagation velocity of the electromagnetic wave within the tumuli. However, based on the average values of the propagation velocity in calcareous stones (and considering that the season was dry and that many void spaces were present among the stones), we can estimate an order of magnitude expected for the depth of the main anomalies visible in Figure 5AC. In particular, by estimating an average relative permittivity of the order of 4 or slightly more, it is easy to estimate that the main outlined anomaly in Figure 5A (tumulus $T_1$) should lie at the depth (from the top of the tumulus) of about 90 cm, whereas the anomaly outlined in Figure 5C (tumulus $T_3$) should lie at a depth of the order of 60 cm from the top of the tumulus. Under tumulus $T_2$, instead, we did not detect any clear anomaly ascribable to a possible tomb. The reasons can be more than one. It is possible that this tumulus is not really an ancient tumulus but something posterior, but it is also possible that an ancient burial structure under the tumulus is completely collapsed or has been destroyed by the roots of the vegetation grown up on the tumulus. It is also to be considered that the vegetation prevented us from crossing tumulus $T_2$ entirely during the prospecting.

## 5. Results Achieved on the Soil

In this section, the results of a conventional GPR investigation in an area close to tumulus $T_1$ are shown. The soil was not smooth enough near the other tumuli, therefore, we avoided further prospecting because the relative results would not have been reliable. The present GPR prospecting is a preliminary task in order to establish whether it is worth plan-

ning a larger geophysical work in the future, according to the achieved results and, above all, to the ground truth that hopefully will be retrieved with some localized excavations.

The GPR system exploited for this was the same system as one exploited for crossing the three tumuli, but now we have used the customary metric wheel of the instrument. In particular, the spatial step of the data (set by the metric wheel) was 1.76 cm for the data at 600 MHz and 3.52 cm for the data at 200 MHz. We performed the measurements along an orthogonal grid (interline step 50 cm) of Bscans around tumulus $T_1$, investigating an area of about $8 \times 12$ m, with a meaningful no-fly zone due to the presence of the tumulus and to some asperities of the soil. Here, we show some results achieved with the antenna at 600 MHz because the antennas at 200 MHz provided similar results with an understandably worse resolution and revealed no meaningful anomaly deeper than the penetration levels allowed by the antennas at 600 MHz. The data were processed according to zero timing, background removal, gain vs. depth (linear and exponential) 1D filtering and Kirchhoff migration. The propagation velocity of the electromagnetic wave in the soil exploited for the migration was 13 cm/ns, evaluated on the basis of the diffraction hyperbolas [24]. Then, depth slices were retrieved too [1,7]. Two slices (deemed to be the most meaningful ones) are shown in Figure 6.

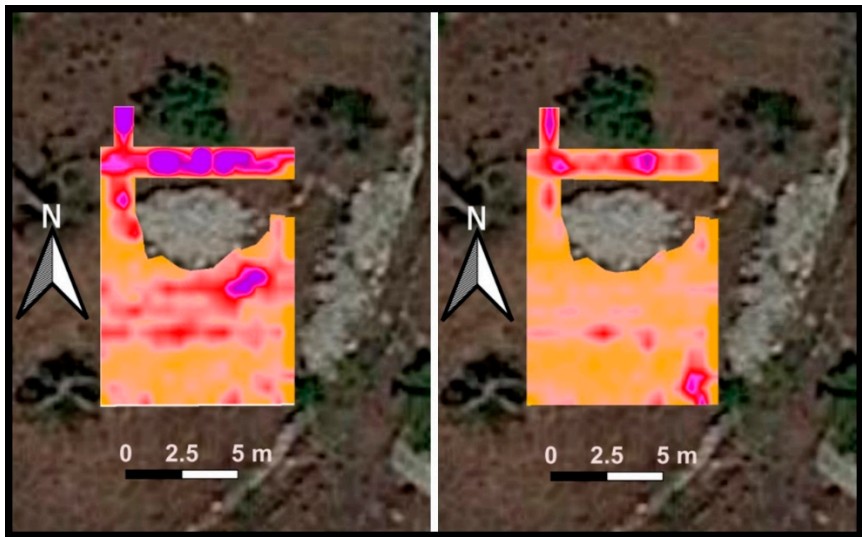

**Figure 6.** Two time slices at 10 (**left**-hand panel) and 20 (**right**-hand panel) ns, approximately corresponding to 65 and 130 cm, respectively. The no-fly zone includes tumulus 1. The false colors represent the intensity of the reflections.

As said, we did not have a differential GPS at disposal, and the slices were georeferenced in QGIS thanks to metric measurements taken in the field, and in particular, thanks to the fact that the measurement lines from the bottom ended at the tumulus.

The results show some anomalies where localized excavations might be carried out. Of course, all the identified anomalies might be also of geological nature with no archaeological meaning. It could be of some interest to also investigate why the reflections behind the tumulus (higher part of the images) are much stronger than the reflection on the other side of tumulus $T_1$ (lower part of the images). In particular, in Figure 7, we also propose a synoptical comparison between two Bscans (before and beyond the tumulus) and the two slices shown in Figure 6. The two Bscans on the top of Figure 7 are the same ones repeated twice.. As can be seen, while before the tumulus (with respect to Figure 6) the scenario seems more homogeneous with a few well-distinguishable anomalies, beyond the tumulus, a plethora of shallow reflectors occur, and some more insulated anomalies are visible only at deeper depth levels. The reason why this appearance is not evident to the eye is because the soil does not exhibit any meaningful difference before and after the tumulus.

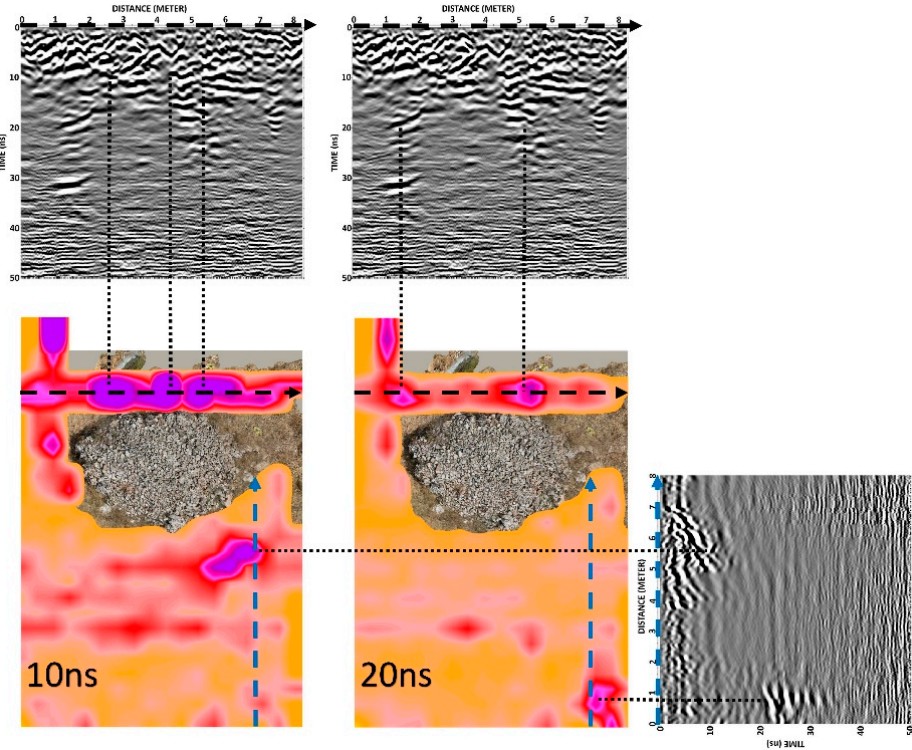

**Figure 7.** Two Bscans whose results are related to the corresponding anomalies visible in the slices of Figure 6.

## 6. Conclusions

In this contribution, we have proposed a GPR exploration in a pre-historic site of cultural interest. In particular, beyond a classical investigation, we performed an investigation of the so-called small *specchie* (tumuli) by prospecting the structures with a GPR system. This required the deactivation of the metric wheel and the consequent activation of the user-stacking acquisition, with the connected use of marker positions. Since only three short Bscans had to be gathered (but with the maximum possible precision with regard to the marker positions), we made use of a method for the acquisition of the data based on the stopover of the instrument at each marker position. This required, in turn, some suitable processing steps in order to account for these stopovers in GPR on the tumuli. The authors are available to provide a video explanation to the readers possibly interested in this acquisition mode. GPR data positions were corrected and integrated by the result of a high-precision photogrammetry survey. As said, this was also performed because we did not have a differential GPS (more precisely, we should say, a differential GNSS, but for simplicity, we will say simply a differential GPS) at disposal. However, independent of this aspect, some further aspects are worth specifying. In particular, a photogrammetric survey can be performed with elementary equipment (even a good mobile phone), whereas GPS equipment is considerably more expensive (several thousand EUR). Moreover, photogrammetric survey can be performed independent of the satellite coverage, which might not be guaranteed due to the presence of shadowing obstacles (in addition, even if it is a rare case, the sensors on the satellites can be switched off at any moment without any notification for military reasons). Nevertheless, GPS enables the measurements of single points, whereas a photogrammetric survey allows for constructing a cloud of hundreds or thousands of points in a short time. In the case at hand, we should also account for the fact that the beam of the GPS takes less comfortable and potentially less precise measurements because it should be taken precisely vertically by the human operator standing on the tumulus. In general, the precision available for the relative height with a GPS and photogrammetric equipment is of the same order (several centimeters [26]). Only a base-rover

GPS system (see, e.g., https://www.topconpositioning.com/gnss/gnss-receivers/hiper-hr#panel-product-specifications, access on 21 January 2022) can achieve sub-centimetric accuracy, but its cost is of the order of EUR 20,000.

The achieved results suggest a probable presence of structures of archaeological interest under at least two of the three tumuli. This work has shown that the investigated area is promising and might reveal relevant prehistoric remains, which could contribute to improving our knowledge of those ancient peoples who used to live in the Salento area during the Bronze Age and, in particular, our knowledge of their burial practices.

**Author Contributions:** Conceptualization, R.P., C.G., S.D.; methodology, R.P., E.C., S.D.; software, R.P. E.C., S.D.; validation, T.Z., E.C.; formal analysis, C.G., R.P.; investigation, T.Z., E.C.; resources, S.D.; data curation, E.C., S.D., R.P.; writing—original draft preparation, R.P., E.C., S.D.; writing—review and editing, C.G., Sebastiano D'Amico; visualization, E.C., T.Z.; supervision, C.G. All authors have read and agreed to the published version of the manuscript.

**Funding:** This research received no external funding.

**Data Availability Statement:** The data will be made available on request.

**Acknowledgments:** This study was supported by an STSM Grant from Cos Action SAGA: The Soil Science & Archaeo-Geophysics Alliance-CA17131 (www.saga-cost.eu, access on 21 January 2022), supported by COST (European Cooperation in Science and Technology www.cost.eu, access on 21 January 2022).

**Conflicts of Interest:** The authors declare no conflict of interest.

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
