# Peer review of "Ground-Penetrating Radar and Photogrammetric Investigation on Prehistoric Tumuli at Parabita (Lecce, Italy) Performed with an Unconventional Use of the Position Markers"

_remotesensing, doi:10.3390/rs14051280_

Round 1

Reviewer 1 Report

Dear authors, I would make the several following comments:

Part 1 (and Part 3):

  • The authors develop in a lengthy manner a way to overcome the problem of not rolling a GPR system over a tumulus. I am wondering why they do not proceed in a step-by-step method (instead of the stop-over procedure), far simpler method considering that the profiles are nor so long over tumuli T1 to T3 (no more than a few meters each.

Part 3:

  • With respect to the sentence “As often happens, raw data are not easily readable”: fortunately, yes in most cases!

RESULTS:

  • “After applying the topographic correction”: simple push-pull along traces, without any attempt to correct for ray inclination.
  • Figure 5: 5B and 5C in text (instead of 5A-T2 and 5B-T3).
  • “have note been migrated”: topographic migration could have been applied.
  • Permittivity of 4 is a theoretical value for “very” dry calcareous stone. Why not having measure velocity with CMP or using diffraction hyperbole?

Part 4:

  • Time slices are not very convincing and this part should be developed.
  • 50cm interline certainly gives under-sampling at 600 MHz frequency. The value should be reduced at least down to 20 cm or less.
  • 13 cm/ns is mere approximation with the previous given permittivity (4).

Author Response

Reviewer n. 1

Dear authors, I would make the several following comments:

Part 1 (and Part 3):

  • The authors develop in a lengthy manner a way to overcome the problem of not rolling a GPR system over a tumulus. I am wondering why they do not proceed in a step-by-step method (instead of the stop-over procedure), far simpler method considering that the profiles are nor so long over tumuli T1 to T3 (no more than a few meters each.

Answer

Thank you for the comments and suggestions, and also for the stressed criticisms throughout the revision. Indeed, to take the data step by step is more easy theoretically, but would require a much longer time in the case at hand. In fact, please note that we have set the spatial step (after interpolation) to 1 cm, and we have gathered an amount of data sufficient to operate such an interpolation. We have comprehensively gathered data along 10.5 linear meters (4.5+3.5+2.5) along the three tumuli. In order to replace this procedure with a step by step measurement with the same spatial step, we should have gathered 1051 punctual measurements. We have gathered the data along the three tumuli in less than 10 minutes: of course the step by step measurements with the same spatial step would have required a much longer time.

Part 3:

  • With respect to the sentence “As often happens, raw data are not easily readable”: fortunately, yes in most cases!

Answer

We have replaced with “As can be seen, raw data are not easily readable”, so referring the statement just to the case at hand.

RESULTS:

  • “After applying the topographic correction”: simple push-pull along traces, without any attempt to correct for ray inclination.
  • Figure 5: 5B and 5C in text (instead of 5A-T2 and 5B-T3).
  • “have note been migrated”: topographic migration could have been applied.
  • Permittivity of 4 is a theoretical value for “very” dry calcareous stone. Why not having measure velocity with CMP or using diffraction hyperbole?

Answers

  • We have inserted the following footnote : “We did not have the possibility to account for the tilting of the antennas while moving long the tumulus”. In particular, we did not have at disposal the GPRSlice or other another code able to account for the tilting of the antennas, i.e. for the ray inclination. We have used (because we had just it at disposal), the reflexw code, that does not allow this option.
  • We have corrected, thanks.
  • The common migration (either FK or in time domain) is a procedure intrinsically based on the hypothesis of a flat interface. The curvature rays of the measurement lines on the tumuli was not large enough to approximate the Bscan as flat (indeed we have tried also a migration with poor results). A migration adapted to curved measurement lines might be conceived, but this is by far beyond the purposes of this paper.
  • The CMP requires separatable antennas or alternatively an array, that moreover should not be “rigid” in order to be displaced continuously on the curved measurement line. We did not have such an equipment at disposal. Moreover, and above all, also the classic CMP procedure requires a flat interface of course, otherwise obvious distortions of the diffraction curves (or lines) occur.

Part 4:

  • Time slices are not very convincing and this part should be developed.
  • 50cm interline certainly gives under-sampling at 600 MHz frequency. The value should be reduced at least down to 20 cm or less.
  • 13 cm/ns is mere approximation with the previous given permittivity (4).

Answers

  • We have added two Bscans compared with the slices (Fig. 7) to offer a more complete and clear dealing.
  • It is true that the spatial Nyquist rate would require an interline step of the order of one fourth of the internal central length, which in the given range of frequencies is a quantity for sure smaller than 50 cm. However, it is also common for practical reasons to gather data with this transect (or even one meter) when exploiting a classical GPR instrumentation not equipped with an array of antennas. The purpose of the prospecting in this about rectangular area was just to perform a preliminary analysis in order to test whether it was worth planning a larger and finer geophysical prospection. We have in particular added the following sentence in Section 4: The present GPR prospecting is a preliminary work in order to establish if it is worth planning a larger geophysical work in the future, according to the achieved results and above all to the ground truth that hopefully will be retrieved with some localised excavations if they will be possible.
  • 13 cm /ns is the velocity estimated on the basis of the GPR data gathered on the soil. It is not related with the velocity hypothesized within the tumulus because the materials are completely different.

Reviewer 2 Report

Dear authors,

the article is written in a good English and is easy to read.
The structure of the paper is flawless and well structured. The method is presented in an excellent way and its arguments are comprehensible.

The article deals with a very interesting improvement of handheld ground penetrating radar. Especially for teams working with manual measurement systems, a topographic correction of the data is an excellent improvement of the survey. The presented method of documenting the terrain by photogrammetry represents a highly accurate and modern recording and should definitely be firmly established in the scientific community. From this point of view, the article is absolutely suitable for publication.

The only shortcoming that can be found in the article is that it only deals with the regional situation in southern Italy. Other research teams in Europe are already working with motorized measurement systems, satellite-based positioning and multi-channel ground radar systems. Nevertheless, these are not a basic requirement for good research.

Therefore, I think that for smaller research teams with limited technical resources, the article could be a significant improvement in their data quality.

In this respect, I would consider the article in its current form suitable for publication.

with best regards

Author Response

Reviewer n. 2

Dear authors,

the article is written in a good English and is easy to read.
The structure of the paper is flawless and well structured. The method is presented in an excellent way and its arguments are comprehensible.

The article deals with a very interesting improvement of handheld ground penetrating radar. Especially for teams working with manual measurement systems, a topographic correction of the data is an excellent improvement of the survey. The presented method of documenting the terrain by photogrammetry represents a highly accurate and modern recording and should definitely be firmly established in the scientific community. From this point of view, the article is absolutely suitable for publication.

The only shortcoming that can be found in the article is that it only deals with the regional situation in southern Italy. Other research teams in Europe are already working with motorized measurement systems, satellite-based positioning and multi-channel ground radar systems. Nevertheless, these are not a basic requirement for good research.

Therefore, I think that for smaller research teams with limited technical resources, the article could be a significant improvement in their data quality.

In this respect, I would consider the article in its current form suitable for publication.

with best regards

Answer

Thank you for the comments and the suggestions. We are aware that there are teams that have achieved interesting very results all over Europe, as e.g. that of the Ludwig Boltzmann Institute for Archaeology. The case at hand was of course a small scale case history and, with specifical regard to the tumuli, a motorized equipment would be heavy and unpractical. Our hope is to have the possibility to perform large scale prospecting and above all archaeological excavations. Indeed, the results of this work have been already sent to the local Superintendence in order to asking permissions for archaeological excavations.

Reviewer 3 Report

Dear Editor,

In this study, Persico et al. present the results of an investigation about the inner structure of the Prehistoric Tumuli or “Specchie” based on measurements from Photogrammetric Digital Elevation models and Ground Penetrating Radar. The study primarily presents the experimental setup employed to apply the two observation strategies. In the second section of the manuscript, the authors report their findings and indicate the suitability of their approach to supporting archeological studies.

In my opinion, the topic treated in the study is of technical interest and fits the scope of a Journal like Remote Sensing. However, I can’t recommend the manuscript for publication, at least in the present form. 

I will summarize my main observations below:

  1. Methodology - the manuscript does not provide enough details about the proposed experimental strategy. Instead, the two approaches are mainly summarized and described qualitatively. 
    1. Let’s consider, for example, the schematic presented in Figure 2. Some details in the graph are not explained either in the figure caption or in the main text. Are details relative to  “Building dense point cloud” and “Building Mesh” described by fundamental literature? If so, please include the relative reference in the main text or present them with further details. 
    2. The same observation applies to the technical instrumentation used to perform the measurements. The two sensors' main characteristics should be available to the readers (consider adding a table or an appendix section).
    3. Consider adding more details regarding the used terminology. Terms like a-scan and b-scan are used without being previously defined.
    4. As mentioned in the introduction, one of the paper's goals is to propose an innovative data acquisition method. Standard methodologies should be mentioned to achieve this objective.
    5. All the information mentioned above should be available to the readers to allow for the reproducibility of the experimental setup.
  2. Presentation of the results - the authors limit their discussion to the presentation of their observations. The authors do not attempt to validate the results or provide any form of uncertainty evaluation. The analysis of remotely sensed data should be either validated using ground observations or provided with an adequate assessment of the relative uncertainty. Without these conditions, the claim that “final results suggest the probable presence of structures of archaeological interest” remains not only unproven but also of no scientific value.
  3. English - the English language and style in the text are not suitable for publication in an international journal. I went through the manuscript several times, and I found several typos and misuse of English words.

Minor comments:

  • All the figure captions should provide a more detailed description of the figure content.
  • Section 1 - report the actual geographic coordinates of the three tumuli.
  • PC - do you mean personal computer. Please define this acronym.
  • Section 2 - the archeological context of the Tumuli is not really relevant here. Please, consider shortening or removing this section and leaving some details to the introduction.
  • Section 2 - “without the addiction of earth” - consider using a different word here. Earth is usually used to refer to the planet.
  • Figure 5 - Include a description of what is indicated by the red-dashed boxes in the figure caption.
  • Figure 6 - What do the colors used in this figure represent? Please, add a description or a color bar.

If the authors decide to submit a new version of the manuscript, I recommend using the correct template for a draft submission (The one with the numbered lines). This would make it easier for the reviewer to provide comments.

Author Response

Reviewer n. 3

In the reviewed article, the authors describe how to use photogrammetry and GPR to search for prehistoric tombs. General note - both methods are very well known and described in the literature. Both methods are used in the study of prehistoric objects.

And the question is: why the known methods and those described in the literature are a reason for publication in a scientific journal? What's revealing here, or the non-use of GPR radar in Southern Italy?

This study is very interesting but for publication in Heritage or Histories (MDPI)

Answer

Thank you for you comments and suggestions. It is true of course that there are many case histories where GPR, photogrammetry and also other techniques have been exploited together. To the best of our knowledge, however, the stop-and-go procedure for gathering GPR data in auto-stacking mode was never published before, and this work is just the first paper proposing and showing the use of this strategy in the field.

Reviewer 4 Report

In the reviewed article, the authors describe how to use photogrammetry and GPR to search for prehistoric tombs. General note - both methods are very well known and described in the literature. Both methods are used in the study of prehistoric objects.

And the question is: why the known methods and those described in the literature are a reason for publication in a scientific journal? What's revealing here, or the non-use of GPR radar in Southern Italy?

This study is very interesting but for publication in Heritage or Histories (MDPI)

Author Response

Dear Editor,

In this study, Persico et al. present the results of an investigation about the inner structure of the Prehistoric Tumuli or “Specchie” based on measurements from Photogrammetric Digital Elevation models and Ground Penetrating Radar. The study primarily presents the experimental setup employed to apply the two observation strategies. In the second section of the manuscript, the authors report their findings and indicate the suitability of their approach to supporting archeological studies.

In my opinion, the topic treated in the study is of technical interest and fits the scope of a Journal like Remote Sensing. However, I can’t recommend the manuscript for publication, at least in the present form. 

I will summarize my main observations below:

  1. Methodology - the manuscript does not provide enough details about the proposed experimental strategy. Instead, the two approaches are mainly summarized and described qualitatively. 
    1. Let’s consider, for example, the schematic presented in Figure 2. Some details in the graph are not explained either in the figure caption or in the main text. Are details relative to  “Building dense point cloud” and “Building Mesh” described by fundamental literature? If so, please include the relative reference in the main text or present them with further details. 

Answer

We have enhanced the text adding the following part:” The photogrammetric processing followed is the typical pipeline which involves a series of consecutive steps as indicated in  Fig. 2 [20-22]. The first step of the photogrammetric workflow is the images import into the photogram-metric software where they will be processed. The following step, called camera alignment, allows for the automatic orientation of cameras and images in space. A sparse point cloud is then created, which forms a 3D point cloud with scattered points. In the next phase, the low-density point cloud is thickened by increasing the number of points and a dense point cloud is generated and, starting from this, a continuous surface is reconstructed made up of polygons whose vertices are the points of the dense cloud. The previous step is called mesh reconstruction and, once the texture is applied to this model, the result is a textured 3D model.”

    1. The same observation applies to the technical instrumentation used to perform the measurements. The two sensors' main characteristics should be available to the readers (consider adding a table or an appendix section).

Answer

With regard to the photogrammetry, we have added a table (Table 1) with the main characteristics of the instrument.

With regard to the GPR data, we have been more precise on the identification of the measurement parameters adding more details. I particular, we have added the following sentences for the measurements on the tumuli: “The GPR was set with a time step of 0.125 ns for the antennas at 600 MHz and 0.25 ns for the antennas at 200 MHz (with 1024 time samples in both cases). User driven staking modality was chosen (not all the systems allow it, but the Ris Hi-Mode does), and a slowing factor 10 was assigned to the launching of the time pulses (this corresponds to fix the time interval between the transmission of two consecutive electromagnetic pulses). We chose this on a heuristic basis, looking at the velocity with which the screen was progressively filled up.”

And we have added the following sentence with regard to the measurements on the earth (with the odometer) close to Tumulus n. 1:

In particular, the spatial step, set by the odometer, of the data, was 1.76 cm for the measurements gathered with the antennas at 600 MHz and 3.52 cm for the data gathered at 200 MHz.”

We have also added the web site of IDSGeoradar s.r.l., which is the manufacturer of the GPR exploited in the present case history.

    1. Consider adding more details regarding the used terminology. Terms like a-scan and b-scan are used without being previously defined.

Answer

We have defined the Ascan and the Bscan in a more precise way, in particular adding enhancing sentences where these two terms were quoted for the first time.

    1. As mentioned in the introduction, one of the paper's goals is to propose an innovative data acquisition method. Standard methodologies should be mentioned to achieve this objective.

Answer

In the introduction, we have added the following sentence: “In particular, we have made use of the K2 FASTWAVE acquisition software, but to our knowledge any code for the acquisition of the data, if equipped for the autostaking acquisition mode, works in the same way.”

Indeed, the classical modality of acquisition in autostaking mode is described just before. Of course, being something basic, it is difficult to find a paper that describes the procedure, but we have done just because we have proposed something different from the common praxis in autostaking mode. 

    1. All the information mentioned above should be available to the readers to allow for the reproducibility of the experimental setup.

Answer

We hope that the insertions of the previously listed sentences in the text enables any reader to reproduce the measurement. Indeed the set-up is just a commercial GPR system, what is innovative (at least in our opinion) is the proposed procedure for the acquisition of the data, that of course is suitable in particular situations as that at hand or similar, and not for the “daily” use of the instrument. Moreover, we have added the following sentence in the conclusion:

“The authors are available to providing a video explanation to the readers possibly interested in this acquisition mode”.

  1. Presentation of the results - the authors limit their discussion to the presentation of their observations. The authors do not attempt to validate the results or provide any form of uncertainty evaluation. The analysis of remotely sensed data should be either validated using ground observations or provided with an adequate assessment of the relative uncertainty. Without these conditions, the claim that “final results suggest the probable presence of structures of archaeological interest” remains not only unproven but also of no scientific value.

Answer

Please note that GPR prospecting is not a technique that allows an “electronic” repetition of the measurements, as instead other techniques do (for example the geoelectrics, where there are fixed electrodes that can gather measurements in the same point N times). In order to have N GPR measurements you should just repeat the measurement campaign N times. This is of course unpractical, and moreover the uncertainty retrieved (e.g. the variance of each value of the gathered signal) would depend not only on the instrument in itself but also on the different paths run by the human operator each time and on the change of the conditions of the soil through the N measurement campaigns, which might be not negligible at all. Of course, the instruments does some internal average (the staking is also this in the end) of the gathered data, but this process is in general internal to the instrument and essentially “dark” for the user.

  1. English - the English language and style in the text are not suitable for publication in an international journal. I went through the manuscript several times, and I found several typos and misuse of English words.

Answer

We have re-read the paper carefully, and hopefully we have decreased substantially the number of misprints and not fluent expressions.

Minor comments:

  • All the figure captions should provide a more detailed description of the figure content.

Answer

We have enhanced the captions of Figs. 1,4, 5 and 6. In the new version a further figure appears too (Fig. 7) absent in the previous version.

  • Section 1 - report the actual geographic coordinates of the three tumuli.

Answer

We have done, in particular the geographic coordinates have been inserted in the caption of Fig. 1.

  • PC - do you mean personal computer. Please define this acronym.

Answer

We have done

  • Section 2 - the archeological context of the Tumuli is not really relevant here. Please, consider shortening or removing this section and leaving some details to the introduction.

Answer

Indeed, this case history was born because of the tumuli and was conceived as a method to investigate the tumuli. So, it is important to understand why tumuli are important for this work, and we hope that further archaeologists that might apply this method for further tumuli. In particular, if the paper is published we will sent it to some archaeologists working in Italy but outside the Salento region.

  • Section 2 - “without the addiction of earth” - consider using a different word here. Earth is usually used to refer to the planet.

We have replaced “earth” with “terrain”.

  • Figure 5 - Include a description of what is indicated by the red-dashed boxes in the figure caption.

We have done

  • Figure 6 - What do the colors used in this figure represent? Please, add a description or a color bar.

We have done

If the authors decide to submit a new version of the manuscript, I recommend using the correct template for a draft submission (The one with the numbered lines). This would make it easier for the reviewer to provide comments.

We have done

Round 2

Reviewer 3 Report

I would like to thank the authors for submitting a revised version of the article. I think that, overall, the manuscript has improved with respect to the previous version. As stated in my first review, I find the topic and the experimental results presented in the study of technical interest suitable for publication in a journal like Remote Sensing.

My main comments and suggestions were focused on the presentation of the work done and the analysis of the results. I think that the authors fulfilled my recommendations only in part. In my opinion, further improvements are needed before I can recommend the study for publication.

In a revised version of the article, the authors should further improve the manuscript in terms of:

  • Presentation of Methodology and Results - the manuscript remains rich in non-necessary qualitative considerations. It would surely benefit from some more quantitative expression of results and the related uncertainties.
  • English - English language and style in the text are STILL not suitable for publication in an international journal. I am a native Italian speaker, and I can see that a good portion of the main text results from a literal translation from Italian to English. It honestly looks like an automatic tool like Google Translate was used.

I won’t have particular problems recommending the article for publication once the authors have fulfilled these recommendations.

Author Response

Reviewer n. 3

I would like to thank the authors for submitting a revised version of the article. I think that, overall, the manuscript has improved with respect to the previous version. As stated in my first review, I find the topic and the experimental results presented in the study of technical interest suitable for publication in a journal like Remote Sensing.

My main comments and suggestions were focused on the presentation of the work done and the analysis of the results. I think that the authors fulfilled my recommendations only in part. In my opinion, further improvements are needed before I can recommend the study for publication.

In a revised version of the article, the authors should further improve the manuscript in terms of:

  • Presentation of Methodology and Results - the manuscript remains rich in non-necessary qualitative considerations. It would surely benefit from some more quantitative expression of results and the related uncertainties.
  • English - English language and style in the text are STILL not suitable for publication in an international journal. I am a native Italian speaker, and I can see that a good portion of the main text results from a literal translation from Italian to English. It honestly looks like an automatic tool like Google Translate was used. 

I won’t have particular problems recommending the article for publication once the authors have fulfilled these recommendations.

 Answer

We also thank the reviewer for his/her work. We have strongly rephrased the paper, that now should hopefully be less redundant and less verbose. With regard to the uncertainties, we have added the order of magnitude of the discrepancy between the marker positions (as identified along the tape) and the points of stop-over of the centre of the GPR antenna box. This position error is of the order of 2-3 cm. Honestly we did not record all the precise values and so we are not able to provide the standard deviation or more refined statistical parameters.

Reviewer 4 Report

Dear authors, the first reading of this article gave me mixed feelings about the scientific value of the study. Now, with the changes, the text is much better cognitively and scientifically. Unfortunately, I cannot agree that new technologies were used. It is true that in this area, the GPR method for the detection of tombs is new and has not been developed before. The overall impression is optimistic and the study is relatively interesting.

However, I still miss the link between the photogrammetric measurements and the GPR. Both measurements are discussed in detail separately, even I have the impression that they are too accurate described.

Why was the photogrammetric method used? Wouldn't it be better to measure the location of the GPR using the GPS method?

In the summary, there is no information about the necessity to use the photogrammetric method and its superiority over other methods.

After the necessary corrections in the Summary and the justification of the surface measurement method used (photogrammetry), the article can be published.

Author Response

Reviewer n. 4

Dear authors, the first reading of this article gave me mixed feelings about the scientific value of the study. Now, with the changes, the text is much better cognitively and scientifically. Unfortunately, I cannot agree that new technologies were used. It is true that in this area, the GPR method for the detection of tombs is new and has not been developed before. The overall impression is optimistic and the study is relatively interesting.

However, I still miss the link between the photogrammetric measurements and the GPR. Both measurements are discussed in detail separately, even I have the impression that they are too accurate described.

Why was the photogrammetric method used? Wouldn't it be better to measure the location of the GPR using the GPS method?

In the summary, there is no information about the necessity to use the photogrammetric method and its superiority over other methods.

After the necessary corrections in the Summary and the justification of the surface measurement method used (photogrammetry), the article can be published.

Answer

We thank the reviewer for his/her appreciation. It is true that no new technology has been created or proposed, because both the GPR system and camera exploited for the photogrammetry were commercial instruments, and also the software for the processing were commercial codes. But honestly we have never stated that we have introduced a new technology. The innovativeness, if any, was referred to the methodology, i.e. we have exploited in a new way a technology already existent and assessed. We might say that we have proposed a new technique but not a new technology. At any rate, we have replaced the word “innovative”, that can drive the reader to think of a new hardware, with the word “unconventional”. In particular, also the title has been changed accordingly.

With regard to the photogrammetry and the GPS, indeed we did not have a differential GPS at disposal and in the last version we have specified this. In particular, also the measurements on the soil have been georeferenced on the basis of metric measurements in the field (also this has been specified in the last version).

In the conclusions we have more precisely specified the reasons of the photogrammetric survey, inserting the following paragraph:

“As said, this was performed also because we did not have a differential GPS (more precisely, we should say a differential GNSS, but for simplicity we will say just a differential GPS) at disposal. However, independently from this aspect, some further aspects are worth specifying. In particular, a photogrammetric survey can be performed with an elementary equipment (even a good mobile phone), whereas a GPS equipment is quite more expensive (several thousand euros). Moreover, a photogrammetric survey can be performed independently from the satellite coverage, that might not be guaranteed due to the presence of shadowing obstacles (in addition, even if it is a rare case, the sensors on the satellites can be switched off in any moment without any notification for military reasons). Still, a GPS enables to the measurements of single points whereas a photogrammetric survey allows to construct a cloud of hundreds or thousands points in a short time. In a case at hand, we should account also for the fact that the beam of the GPS makes less comfortable and potentially less precise the measurements, because it should be taken precisely vertical by the human operator standing on the tumulus. In general the precision available for the relative height with a GPS and a photogrammetric equipment is of the same order (several centimetres): only a base-rover GPS system (see e.g.  https://www.topconpositioning.com/gnss/gnss-receivers/hiper-hr#panel-product-specifications) can achieve sub-centimetric accuracy, but its cost is of the order of 20000 euros.”